# Quick, Coordinated and Authentic Reprogramming of Ribosome Biogenesis during iPSC Reprogramming

**DOI:** 10.3390/cells9112484

**Published:** 2020-11-15

**Authors:** Kejin Hu

**Affiliations:** Department of Biochemistry and Molecular Genetics, School of Medicine, University of Alabama at Birmingham, Birmingham, AL 35294, USA; kejinhu@uab.edu

**Keywords:** ribosome biogenesis, mesenchymal-to-epithelial transition (MET), fibroblasts, human embryonic stem cell (ESC), reprogramome, reprogramming legitimacy, induced pluripotent stem cell (iPSC), RNA-seq, OCT4/SOX2/KLF4/MYC (OSKM), transcriptome, transcription profiling

## Abstract

Induction of pluripotent stem cells (iPSC) by OCT4 (octamer-binding transcription factor 4), SOX2 (SR box 2), KLF4 (Krüppel-Like Factor 4), and MYC (cellular Myelocytomatosis, c-MYC or MYC) (collectively OSKM) is revolutionary, but very inefficient, slow, and stochastic. It is unknown as to what underlies the potency aspect of the multi-step, multi-pathway, and inefficient iPSC reprogramming. Mesenchymal-to-epithelial (MET) transition is known as the earliest pathway reprogrammed. Using the recently established concepts of reprogramome and reprogramming legitimacy, the author first demonstrated that ribosome biogenesis (RB) is globally enriched in terms of human embryonic stem cells in comparison with fibroblasts, the popular starting cells of pluripotency reprogramming. It is then shown that the RB network was reprogrammed quickly in a coordinated fashion. Human iPSCs also demonstrated a more robust ribosome biogenesis. The quick and global reprogramming of ribosome biogenesis was also observed in an independent fibroblast line from a different donor. This study additionally demonstrated that MET did not initiate substantially at the time of proper RB reprogramming. This quick, coordinated and authentic RB reprogramming to the more robust pluripotent state by the OSKM reprogramming factors dramatically contrasts the overall low efficiency and long latency of iPSC reprogramming, and aligns well with the potency aspect of the inefficient OSKM reprogramming.

## 1. Introduction

Human pluripotent stem cells (PSCs) can be induced by ectopic expression of four factors, OCT4, SOX2, KLF4, and MYC (collectively OSKM) in somatic cells, most commonly fibroblasts [1,2,3]. Induction of PSCs (iPSCs) from human fibroblasts is still very inefficient, slow, and stochastic [4,5,6,7]. Our understanding about the molecular events underlying the mixed outcomes of iPSC generation is very rudimentary. Much research has been conducted to understand the molecular mechanisms of OSKM reprogramming, but is compromised by low efficiency [8]. Transcriptional and chromatin-binding data were collected from reprogramming cells, intermediate partially reprogrammed cell lines, and the established iPSC lines [9,10,11,12]. These analyses implicitly treated all these transcriptional and binding data as positive responses, partly because scientists were so amazed by this revolutionary technology and apparently ignored the fact that 99% of the data represent the cells that do not go in the direction of pluripotency. Their purified partially reprogrammed cells were also heterogenous with limited potentials to be further reprogrammed [10].

To attenuate the issues of reprogramming noises in dissecting the molecular underpinnings and limitations of OSKM reprogramming, the author recently proposed a new concept of reprogramome [13]. Reprogramome is the complete set of genes to be reprogrammed. Reprogramome includes upreprogramome and downreprogramome. Upreprogramome is the full subset of genes that should be upregulated to the levels found in PSCs, and downreprogramome is the subset of genes that should be downregulated to the levels found in PSCs. The concept of reprogramome provides a foundation for evaluation of the reprogramming legitimacy of transcriptional responses of a gene to the OSKM factors [8]. On the basis of the concept of reprogramome and reprogramming legitimacy of transcriptional response of a gene to the reprogramming factors, the author identified a pattern of transcriptional responses to OSKM reprogramming, i.e., the PIANO responses, standing for Proper, Insufficient, Aberrant, and NO reprogramming at the initial stages of reprogramming [8]. The PIANO responses to reprogramming include positive, negative, and refractory reprogramming. This mixed outcomes effectively explain the robustness as well as the limitations of OSKM reprogramming. The concepts of reprogramome and reprogramming legitimacy provide a more logical avenue to dissecting the reprogramming.

The mammalian ribosome is a complex supramolecular assembly of 4 major ribosomal ribonucleic acids (rRNA) and 80 ribosomal proteins (RP) as a protein translation machinery essential for every cell [14]. The mature functional ribosome contains two subunits, 40S small subunit (SSU) and 60S large subunit (LSU). SSU comprises the mature 18S rRNA and 33 RPs, while the LSU consists of 5S, 5.8S, and 28S rRNAs associated with 47 RPs. Ribosome biogenesis (RB) represents the most active and energy-consuming cellular process. In each second there are an estimated 40 ribosomes produced in a growing yeast cell [15], and 125 ribosomes generated in a growing HeLa cell [16]. In yeast, 65–70% of global transcription and 30% of the global translation are dedicated to RB [17]. RB is also extremely complex but highly coordinated involving syntheses of rRNA and RP-coding mRNA by all the three polymerases (Pol I, Pol II, and Pol III), extensive rRNA processing and modifications, ribosome subunit assembly with the support of ribosome biogenesis factors (RBF), exporting of the pre-40S and pre-60S subunits into the cytoplasm, maturation of the ribosome subunits, and assembly of the functional mature 80S ribosomes in the cytoplasm. The mammalian polycistronic 13-kb 47S pre-rRNA is transcribed from the repetitive ribosomal DNA (rDNA) arrays in nucleoli by Pol I [18]. The resulting 47S rRNA transcript is processed extensively but through highly ordered multiple steps by the endonucleases and exonucleases in nucleoli and nucleoplasm, as well as in cytoplasm to remove the external transcribed spacers and internal transcribed spacers, eventually generating the mature 18S, 5.8S, and 28S species [19]. At the same time, 5S rRNA is transcribed in the nucleoplasm by Pol III. One critical RB process is the extensive modifications of rRNAs including ribose methylation, pseudouridylation, and base methylation guided by small nucleolar ribonucleoproteins (snoRNP) [20]. More than 200 sites in rRNAs, most of them being in the coding regions, are covalently modified. Genes for RPs and RBFs are transcribed by Pol II in the nucleoplasm, and translated by ribosome itself in cytoplasm, and then imported back into the nucleus for assembling of the ribosome subunits. The rRNA and RPs then assemble in multiple-step processes with the support of RBFs. The assembled export-competent LSU and SSU are then exported actively by the nuclear pore complex into the cytoplasm to further mature and assemble the functional 80S ribosomes. The highly complex and tightly regulated process of RB is well studied in budding yeast, but is much less clear in human cells [21].

Once considered as a ubiquitous and house-keeping pathway, RB has through accumulating evidence been indicated as playing tissue- and cell type-specific roles. Defects in RB results in cancers [14] and various other diseases, generally known as ribosomopathies such as Diamond–Blackfan anemia, Schwachman–Diamond syndrome, and others. Defects in the ubiquitous and essential RB pathways should affect all tissues and cell types, but the paradox is that the ribosomopathies are very tissue-specific [22]. It is now clear that RB varies among cell types and that the concentration of ribosomes ranges from 1 to 10 million per cell. Some genes of RB are shown to be critical in regulation of pluripotency and reprogramming [23,24,25,26]. The global expression landscapes of RB-related genes in stem cells remains poorly documented [27]. It is not clear whether there is a difference in ribosome biogenesis between the starting fibroblasts and the end product, iPSCs. If there is indeed a difference, how efficient are the Yamanaka factors in reprogramming the RB pathways into the pluripotent state from the somatic state?

A well-studied natural reprogramming process observed in embryogenesis, tissue repair, and disease development is the epithelial-to-mesenchymal transition (EMT) [28,29]. EMT is the process wherein epithelial cells undergo morphogenetic changes from the laterally adjoined cells with an apico-basal polarity towards more isolated motile mesenchymal cells with a front-end/back-end polarity. EMT occurs many times during various stages of embryogenesis for tissue development and organogenesis, as well as wound healing, fibrosis, and cancer progression. The motile mesenchymal cells then move to new locations. In the new environments, mesenchymal cells may undergo a reverse process called mesenchymal-to-epithelial transition (MET), and the resulting secondary epithelial cells then form new tissues or organs, repair the wounds, develop fibrosis, or develop cancers in new nodules. It is now well established that fibroblast reprogramming towards iPSCs also undergoes an MET process [9,30,31,32]. MET in iPSC generation is an essential early step. It is not known whether any other reprogramming of cellular features precedes MET during iPSC generation.

This report shows that RB was more robust in human PSCs (hPSCs) than in the starting fibroblasts of iPSC reprogramming. Using RNA sequencing (RNA-seq) and the concept of reprogramming legitimacy, this study then demonstrated that RB pathways were properly reprogrammed to the pluripotent state in a highly coordinated manner as early as 48 h post-transduction of OSKM in human fibroblasts. On the basis of the same set of data, it is clear that MET barely initiated at the same time points, and reprogramming of the RB network is a very early event at least 2 days before MET substantially initiates. Additional analyses with four independent iPSC lines and fibroblasts from a different donor led to the same conclusions about the pluripotent ribosome biogenesis and its reprogramming by the OSKM reprogramming factors. The findings here provide additional molecular insights about the robustness of the OSKM reprogramming factors being able to convert a small portion of the transduced fibroblasts into the pluripotent state.

## 2. Results

### 2.1. More Robust Ribosome Biogenesis (RB) in hESCs than in the Reprogramming Starting Fibroblasts

The author recently reported that ribosome biogenesis (Gene Ontology (GO) #0042254) is among the list of the top GO terms for the genes that have already been properly upreprogrammed to the pluripotent state within 48 h post-transduction of the Yamanaka reprogramming factors. This interesting finding prompted the author to ask the question of whether there is any difference in RB between PSCs and fibroblasts, the latter of which is the widely used starting cell type for iPSC generation. Is RB more robust in hPSCs than in fibroblasts in general? To answer this, the current study first extracted RNA-seq data for the subset of the 298 human genes involved in RB (Appendix A) and compared their transcription levels between human embryonic stem cells (hESCs) and fibroblasts. Indeed, the overall expression levels for the RB gene set were higher in hPSCs than in fibroblasts (Figure 1A). Further analyses indicated that 114 of those genes are expressed at least 2× higher (q < 0.01, hESC = 3, fibroblasts = 4) in hESCs than in fibroblasts (Appendix A), while 96 genes were similarly expressed in both cell types (Appendix A). Markedly, only 11 genes were enriched in fibroblasts on the basis of the same stringency (Appendix A). The enrichment of these genes in fibroblasts may not represent strength of ribosome biogenesis in fibroblasts, and possibly represents negative regulation of ribosome biogenesis or roles in other cellular processes (see the Discussion section).

Considering the biased enrichment for RB genes in hESCs, this study applied less stringent criteria to further compare the expression profiles. It was found that additional 56 genes were expressed at least 1.5-fold higher in hESCs (q < 0.05) (right in Figure 1B), which brought the total number of the hESC-enriched genes to 170 (Figure 1B,C, Appendix A). The author recently proposed a mathematical model to calculate the transcriptional differences for a subset of genes involved in a specific cellular feature/process [13]. Using the formula, it was estimated that the hESC enrichment of ribosome biogenesis was 188.6 log2 fold changes (LFCs) when only the 114 genes were considered, while the fibroblast enrichment was 22 LFCs only (Figure 1D). Therefore, there is a great net RB enrichment in hESCs. The enrichment should be much greater considering that an additional 56 RB genes were also significantly enriched in hESCs, albeit at lower degrees, and that the significant downregulation of the 11 fibroblast-enriched genes in hESCs may also constitute the robustness of ribosome biogenesis in hPSCs as the hPSC-enriched genes do (see the Discussion section).

Of note, seven of the RB genes were not expressed in both cell types using our criteria (*PIH1D2*, *RPLP0P6*, *RPL3L*, *METTL15P1*, *RPS10P5*, *RPL10L*, and *VCX*). It is not surprising in that *RPLP0P6*, *METTL15P1*, and *RPS10P5* are all pseudogenes that are duplicated by retrotransiposition and may become dead genes [33]; RPL3L is a tissue-specific gene with an uncertain functional role in ribosome biogenesis [34]. As a homolog of the ribosomal protein L10 (RPL10), *RPL10L* (ribosomal protein L10 like) is an intronless gene, a hallmark of pseudogene [33,35], and it is expressed in testis only [36]. The nucleolus protein VCX is expressed exclusively in male germ cells only [37]. PIH1D2 may has been wrongly annotated as a ribosome biogenesis GO because of a very similar name with PIH1D1 [38]. Human PIH1D1 and PIH1D2 share very low sequence identity (21%). Only PIH1D1, but not PIH1D2, is a component of the R2TP complex, which has a role in box C/D snoRNP biogenesis [38]. The negative results about these silenced “ribosomal” genes unexpectedly further indicate that our RNA-seq data are very informative and accurate, being able to distinguish those uncertain “ribosomal genes” in our samples. Therefore, in terms of the two cell types in question here, there are only 291 RB genes in query. Another indication for the sensitivity and accuracy of our RNA-seq data is that our data identified that some of the 11 fibroblast-enriched genes were reported to be negative regulators of ribosome biogenesis (see the Discussion section).

In summary, the majority of RB genes were enriched in hPSCs (58.4%), with 33.3% of them being expressed in both cell types at similar levels, but only 3.8% of these genes were found to be enriched in the reprogramming starting fibroblasts that may include some negative regulators of ribosome biogenesis (see the Discussion section). These data indicate that hPSCs have a more robust ribosome biogenesis system.

### 2.2. Ribosome Biogenesis Was Properly Reprogrammed within 48 Hours

The above analyses indicate that there are significant differences in the RB transcription profiles between human fibroblasts and hPSCs, and Yamanaka factors should reprogram the fibroblast RB transcriptional program to that of hPSCs. Since previous research showed that “ribosome biogenesis” (GO #0042254) was among the top GO terms for the gene set that were properly reprogrammed within 48 h [8], the current research further enquired into the degree of reprogramming for the entire RB system. Clustering analyses with the 114 strictly PSC-enriched RB genes showed that this set of genes clustered with hESCs and was no longer similar to the starting fibroblasts upon OSKM reprogramming (individual clustering and heat map not shown). This was true for both time points (48 and 72 h) of OSKM induction. The analyses were then extended to the larger gene set of the 170 PSC-enriched genes at the less stringent 1.5-fold levels. This more complete set of the PSC-enriched genes was again reprogrammed to the levels of pluripotency within 48 h and became dissimilar to the starting fibroblasts (Figure 2A).

After being shown that the PSC-enriched gene set was reprogrammed to the pluripotent state, this study examined the fibroblast-enriched gene set. In contrast, this set of genes as a group remained not reprogrammed since the reprogramming cells at both time points still clustered with the starting fibroblasts and the green fluorescent protein (GFP) controls (Figure 2B). A scrutiny with this small group of genes, however, revealed that two of them were in fact successfully reprogrammed (*PTEN* and *GTF2H5*) (Appendix A). Finally, the entire set of genes in the ribosome biogenesis was examined to find out whether failure of reprogramming for the fibroblast-enriched gene set make the entire set of ribosome biogenesis genes more dissimilar from the pluripotent RB program. Interestingly, the entire set of 298 RB genes still clustered with the pluripotent state for both time points (Figure 2C). In summary, the PSC-enriched state of the RB transcriptional program has been established, although the minor fibroblast-enriched feature has not yet been erased for 9 of the 11 fibroblast-enriched RB genes.

### 2.3. Accurate Reprogramming of the Ribosome Biogenesis

The above clustering analyses were based on the overall expression profiles for the RB sub-reprogramomes. It is not clear whether each individual RB gene was properly reprogrammed. The RNA-seq dataset for the entire upreprogramome of the 170 hPSC-enriched genes was then examined for the reprogramming status (Appendix A). Previous report indicated that there is insufficient reprogramming for some genes [8]. Using stringent criteria (differences between the reprogramming cells and endpoint PSCs become >1.5×, upregulation by at least 1.5×, *p* < 0.05), there were only eight genes that were insufficiently upreprogrammed (Appendix A). However, if less stringent criteria were used (the differences between reprogramming cells and the PSCs were still >2×, upregulation by at least 2×, *p* < 0.01), only one gene was insufficiently reprogrammed (*CDH7*), for which the expression in the reprogramming cells was still 13.3× lower than that in hPSCs. Nevertheless, the group of these eight genes became clustered with the hESC upon OSKM induction (Appendix A), indicating near-complete reprogramming as a group, albeit with some deficit.

The previous research also indicates that some genes did not respond to OSKM induction at the early stages ^8^. A scrutiny of the 170 PSC-enriched RB genes showed that 29 of these RB genes did not respond to OSKM induction (fold changes caused by OSKM was <1.5×; Appendix A). The above results indicate that the majority of these 170 PSC-enriched RB genes were properly reprogrammed, but a small set of these genes were resistant to OSKM reprogramming.

The previous report indicates that Yamanaka factors resulted in six types of aberrant reprogramming at the early stages of reprogramming including wrong up- and down-reprogramming, unwanted up- and downreprogramming, and over up- and down-reprogramming [8]. It is necessary to investigate whether any RB gene is aberrantly reprogrammed, which may be masked by the overall proper reprogramming. With reference to the above analyses, it is clear that there was one gene that underwent wrong upreprogramming for the fibroblast-enriched RB genes (*PWP2*), but there was no over-downreprogramming. The 170 hPSC-enriched genes were then examined. Of note, not a single gene of the 170 ESC-enriched RB genes was significantly downregulated by OSKM. The greatest downwards fold change was only 1.2× (Appendix A). Lack of downregulation for the RB upreprogramome indicates that the directionality of reprogramming is very accurate, and there is no wrong downreprogramming for this group of genes. Only one gene (*WDR3*) was upregulated to the level that is at least twofold higher than that in the endpoint cells, hPSC (2.1 times), but it was not statistically significant (Appendix A), indicating that the over upreprogramming of this single gene (*WDR3*) was marginal.

Two other types of aberrant reprogramming are unwanted down- and upreprogramming for the genes that should not be reprogrammed at all [8]. To find out whether there are these two types of aberrant reprogramming or not, this study examined the data for the 96 shared RB genes. Surprisingly, as seen with the RB sub-upreprogramome of 170 genes, not a single gene in the 96 shared RB genes was downregulated by twofold, and only two genes (*ZFN658* and *YBEY*) were downregulated to the levels wherein their expression levels become twofold lower than that in the hESCs (−2.7 and −2.2×, respectively), but none of them were significant at the q < 0.01 level. Therefore, there was no unwanted downreprogramming among these 96 common RB genes. Only six genes were upregulated to the levels so that the expressions in the reprogramming cells become at least twofold higher than that in the hESCs. However, only two of them (*URB1* and *TFB2M*) were induced by more than twofold when compared to both of the GFP and naïve fibroblast controls, which were statistically significant at the level of 0.05 but not at 0.01. The expression levels of these two genes were only 2× (*TFB2M*) and 2.6× (*URB1*) higher than that in hESCs. Overall, these 96 common RB genes remain unchanged upon OSKM induction, which is in stark contrast to the responses of the 170 hPSC-enriched RB genes to OSKM induction (Appendix A).

In summary, OSKM tended to upregulate the genes of the ribosome biogenesis pathways, even for the four genes that were aberrantly reprogrammed at the marginal levels. OSKM properly upreprogrammed the majority of the 170 hESC-enriched RB genes within 48 h, with only 29 genes irresponsive at the early stages of reprogramming, and at the same time did not affect the expression statuses of the 96 shared RB genes. In conclusion, these results indicate highly accurate and authentic reprogramming of ribosome biogenesis pathways by OSKM, in stark contrast to the overall iPSC reprogramming, which is inefficient, slow, and stochastic.

### 2.4. Mesenchymal-to-Epithelial (MET) Transition Was not Achieved at the Time of Proper RB Reprogramming

The above data indicate that the pluripotency state of ribosome biogenesis was generally reprogrammed within 48 h post OSKM transduction. It is well known that MET is also an early event in iPSC reprogramming [9,31,32]. The findings above prompted an examination of MET status at the same time points using the same dataset. Mesenchymal transcriptional program is generally regulated by the following six transcriptional factors (TF): ZEB1, ZEB2, SNAI1, SNAI2, TWIST1, and TWIST2. As expected, *ZEB1*, *ZEB2*, *SNAI2*, *TWIST1*, and *TWIST2* were found to be expressed in fibroblasts at high levels, but silenced in hESCs (Figure 3A and Appendix A). *SNAI1*, however, was found to be expressed in both cell types at similar levels. The common *SNAI1* remained unchanged after OSKM induction (Figure 3A and Appendix A). A successful MET requires the silence of the fibroblast-specific mesenchymal TFs. Although the five fibroblast mesenchymal TFs were downregulated by 1.3- to 6.1-fold (compare orange and blue bars in Figure 3D), these downregulations were insignificant in terms of the target expression levels because their expressions were still 14.1- to 170-fold higher than the target expression levels (orange bars in Figure 3D, Appendix A). Because these five TFs are considered not expressed in hESCs, this study also calculated fold differences above the expression threshold, which still ranged from 11.8× to 57.4× (grey bars, Figure 3D, Appendix A), indicating that these genes were still robustly expressed in the reprogramming cells. As a result, the reprogramming cells in terms of the expressions of the mesenchymal TFs remained clustered with the starting cells of reprogramming (Figure 3B and Appendix A).

On the other hand, another sign of MET is the activation of the e-cadherin gene cadherin 1 (*CDH1)*. Although *CDH1* is upregulated by 2.7-fold, it was still considered inactive because its expression level was still under the expression threshold (21.8 vs. 50 DESeq2 normalized read counts; Appendix A, Figure 3A and Appendix A).

In summary, the MET process at most only initiated at the time proper RB reprogramming was achieved because the five fibroblast-specific mesenchymal TFs remained expressed at high levels (Figure 3A–C, and Appendix A) and the epithelial marker e-cadherin gene *CDH1* remained inactive (Figure 3A and Appendix A).

### 2.5. Defining the Sub-Reprogramomes of MET for Human Fibroblast Reprogramming to Pluripotency

The above analyses with the seven key MET/EMT factors indicated that MET merely initiated and these genes remained unreprogrammed at the time of proper RB reprogramming. An important question is: what is the reprogramming status for the remaining EMT/MET genes? To answer this, the following fundamental question must first be answered: what is the EMT/MET molecular profiles in terms of cell fate conversion from fibroblasts to hPSCs? The recent tallying indicates that as many as 1184 genes are involved in the EMT/MET processes [39]. EMT and its reverse process MET occur during different stages of development for organogenesis (type I), wound healing and fibrosis (type II), and cancer progression (type III) [40]. Now, it is known that MET is also an essential and early step for pluripotency reprogramming from fibroblasts [30]. The molecular profiles for EMT/MET are very different and are context-dependent [41].

To evaluate the legitimacy of the transcriptional responses of the MET/EMT genes, this study first defined the MET subreprogramome. Of the documented EMT/MET genes, 300 were fibroblast-enriched (Appendix A) (see the Materials and Methods section for criteria), and 213 were hPSC-enriched (Appendix A). Of the 300 fibroblast-enriched genes, 64 were expressed in fibroblasts only (Appendix A), while 208 were expressed in both cell types (Figure 4A,C, and Appendix A). Of the 213 hESC-enriched genes, 67 were expressed exclusively in hESCs (Figure 4A,D, and Appendix A), while 135 were expressed in both cell types (Appendix A). The expression levels were similar in both cell types for 613 genes (Figure 4A,B, Appendix A), of which 304 genes were not expressed in both cell types while 276 were expressed in both cell types (Figure 4B).

On the basis of the above classification, the MET upreprogramome includes 213 genes, while the downreprogramome contains 300 genes, with the two combined constituting the entire MET reprogramome of 513 genes. Within the MET upreprogramome, there is an activatome of 67 genes, and within the MET downreprogramome, there is an erasome of 64 genes. The author also recently developed a mathematical model to quantitate reprogramming. Using this formula, this study estimated that the amount of upreprogramming for MET is 836.9 log2 fold changes (LFC), while the amount of downreprogramming of MET is 1160.6 LFC (Figure 4E). This quantification indicates that, in iPSC generation, erasure of the fibroblast EMT/MET features is more predominant than establishing the pluripotency EMT/MET signature.

### 2.6. None of the Sub-Reprogramome of the MET Gene Set Was Reprogrammed at the Time of Proper Reprogramming of Ribosome Biogenesis

Although the six signature MET genes were not reprogrammed as required at the time of proper RB reprogramming, it is not clear if the MET reprogramome or any of the MET subreprogramomes defined above were properly reprogrammed at the two time points. In contrast to ribosome biogenesis, the MET reprogramome of 513 genes remained clustered, with the starting fibroblasts and the control fibroblasts transduced with GFP viruses at both time points (48 and 72 h) (Figure 5A). The same was true for all the MET subreprogramomes, including upreprogramome, downreprogramome, activatome, and erasome (Figure 5B–E, respectively, Appendix A). All of these subsets remained clustered with the starting fibroblasts and GFP-transduced fibroblasts but not with the hESCs at both time points of reprogramming, indicating that none of these subsets were reprogrammed.

### 2.7. PIANO Responses of the MET/EMT Genes to the Yamanaka Factors at the Time of Proper RB Reprogramming

The MET/EMT genes were further tested for the PIANO responses to reprogramming factors reported before [8]. All types of PIANO responses existed for the MET/EMT genes but the “no response” category dominated (Figure 6 and Appendix A). A total of 104 fibroblast-enriched MET genes were resistant to OSKM reprogramming (Figure 6C and Appendix A), and 99 hESC-enriched MET genes were irresponsive to OSKM reprogramming (Figure 6D and Appendix A). Insufficient reprogramming was the second largest group. A total of 56 fibroblast-enriched MET genes were significantly downregulated by at least twofold, but insufficiently downreprogrammed with a deficit of at least twofold (Figure 6A and Appendix A). Although there were significant dowreprogramming for these 56 fibroblast-enriched genes, this group still clustered with the starting fibroblasts and was dissimilar to hPSCs. At the same time, 11 hPSC-enriched genes were significantly upregulated by at least twofold but were insufficiently upreprogrammed, and this group became more similar to hESCs (Figure 6B and Appendix A). Five common MET genes underwent unwanted downreprogramming (Figure 6E and Appendix A), while 11 common MET genes were upregulated by at least twofold when they should not have been (unwanted upreprogramming) (Figure 6F and Appendix A). Two MET genes were wrongly downregulated by at least twofold where they should have been upreprogrammed by at least twofold (Figure 6G and Appendix A), while five MET genes were wrongly upregulated by at least twofold where they should have been downreprogrammed by at least twofold (Figure 6H and Appendix A). A total of 22 fibroblast-enriched MET genes were properly downreprogrammed to the levels found in hPSCs (Figure 6I and Appendix A), and 17 hESC-enriched genes were properly upreprogrammed to pluripotency levels (Figure 6J and Appendix A). There were three over-upreprogrammed and two over-downreprogrammed genes (data not shown).

In summary, predominant genes of the MET reprogramome did not respond to OSKM reprogramming, and severe under reprogramming for the fibroblast-enriched MET genes was observed. Nevertheless, a small portion of MET genes were properly reprogrammed while all six types of aberrant reprogramming were observed. This mixed outcome for MET reprogramming agrees with the conclusion that MET reprogramming initiates but is not achieved at the time of proper RB reprogramming.

### 2.8. Additional Epithelial Signatures Were Not Reprogrammed at the Time of Proper Reprogramming of Ribosome Biogenesis

Only one epithelial marker e-cadherin gene (*CDH1*) was individually examined for its reprogramming status at the early stages thus far, and the author finally examined the reprogramming statuses of additional annotated epithelial genes among the 67 MET/EMT genes unique to hPSCs. As expected, GO analyses indicated that as many as 31 GO terms with the key words of “epithelial” or “epithelium” were overrepresented by these 67 hESC-specific genes (Appendix A). A total of 28 out of the 67 hESC-specific genes were associated with these 31 epithelial GO terms (Appendix A). Although six genes were associated with the GO terms of “regulation of epithelial to mesenchymal transition”, “regulation of epithelial to mesenchymal transition”, and “mesenchymal cell proliferation”, all of these were associated with other epithelial GO terms except for T-Lymphoma Invasion And Metastasis-Inducing Protein 1 (*TIAM1)*. The literature indicates that TIAM1 promotes the formation of adheren junction and induces epithelial phenotyps [42]. In fact, it is known that some genes play roles both in MET and EMT, for example, the Wilms tumor transcriptional factor WT1 [30]. Like the seven key MET/EMT genes and the MET subreprogramomes, these 28 hESC-specific epithelial genes were not reprogrammed individually nor as a group except for *SOX2*, *F11R*, *TGFA*, and *BMP6* (Figure 7 and Appendix A). It is well known that *SOX2* is not activated at this time point. The high expression level of *SOX2* is from the transgenes. *TGFA* and *BMP6* were properly reprogrammed, but *F11R* was insufficiently upreprogrammed although upregulated significantly. In summary, the 28 hESC-specific genes annotated as the epithelial terms were not reprogrammed, except for two of them at the time of proper reprogramming of ribosome biogenesis.

### 2.9. More Robust Ribosome Biogenesis was Observed in iPSCs

The author also investigated whether human iPSCs displayed more robust ribosome biogenesis. RNA-seq data for four lines of the verified iPSCs [4,5] were compared with that of human fibroblasts. Out of the listed 298 RB genes, 176 were expressed significantly higher in iPSCs than in fibroblasts (q < 0.05, >1.3×) (Figure 8A, Appendix A). Among the 176 iPSC-enriched RBs, 107 were enriched by at least twofold at the more stringent level (q < 0.01) (not separately visualized). Of note, 159 of these iPSC-enriched RB genes were among the 170 RB genes enriched in human ESCs cells. In contrast, only 17 RB genes were expressed significantly higher in fibroblasts than in iPSCs (>1.3×, q < 0.05). When stringent sorting criteria were used (q < 0.01, FC > 2×), only 10 genes were expressed significantly higher in fibroblasts (Figure 8B). In agreement with the ESC data, these 10 iPSC-low RB genes were among the 11 fibroblast-enriched RB genes when compared to ESCs (Figure 8B). Calculation of enrichment for the RB pathway [13] indicated a predominant RB enrichment in iPSCs (Figure 8C). These data indicate that the more robust ribosome biogenesis is a conserved feature between iPSCs and ESCs. We also found the same seven RB genes were not expressed in iPSCs (*PIH1D2*, *RPLP0P6*, *RPL3L*, *METTL15P1*, *RPS10P5*, *RPL10L*, and *VCX*). Some differences between iPSCs and ESCs may represent some minor incomplete or aberrant reprogramming in our iPSC samples, although these lines were tested to be pluripotent [4,5].

### 2.10. Robust Ribosome Biogenesis Remained a Pluripotent Feature When Compared to an Independent Human Fibroblast Line

The author also examined whether the more robust ribosome biogenesis can be observed when compared to fibroblasts from a different individual. Indeed, compared with data from another human fibroblast line (ATCC cell line CCD-1079Sk, CCD fibroblast hereafter), 156 RB genes were significantly enriched in ESCs (>1.3×, q < 0.05, CCD = 4, ESC = 4) (Appendix A, Figure 9A). Of note, 128 of these RB genes were among the list of 170 pluripotency-enriched RB genes obtained using the first set of RNA-seq data. Again, only 34 RB genes demonstrated significantly higher expression in CCD than in ESCs (>1.3×, q < 0.05). Among these 34 fibroblast-enriched gens, only 11 were enriched by at least twofold (q < 0.01) (Figure 9B). The remaining 32 all had lower enrichment (<2×). Of note, this small pool of highly enriched fibroblast RB genes was very consistent (10 out 11 are shared by all three comparisons discussed above). Calculation of expression enrichment using the reported model [13] indicated a predominant enrichment of RB in ESCs (Figure 9C).

### 2.11. OSKM Quickly Upregulated Ribosome Biogenesis to Pluripotent State in An Independent Human Fibroblast Line

The author next tested whether ribosome biogenesis can be globally reprogrammed to pluripotent state in a fibroblast line from a different donor. RNAs from the reprogramming CCD fibroblasts at 48 and 72 h were sequenced. Out of the 289 RB genes, 200 were upregulated by OSKM by at least 1.3-fold (q < 0.01, OSKM = 4, CCD = 4) (Appendix A, Figure 10A). Among these 200 OSKM-upregulated RB genes, 123 were expressed at higher levels in ESCs by at least 1.3-fold (q < 0.05), and these 123 RB genes became clustered with ESCs and moved away from the parental fibroblasts (Figure 10B). Detailed scrutiny indicated that 125 of the 200 OSKM-regulated genes were upregulated by at least twofold. Of note, the fibroblasts expressed none of these 125 genes at higher levels by twofold, and expressed only one gene at higher expression level at the significant level of q < 0.01 (1.4-fold). On the other hand, OSKM downregulated only nine RB genes (Figure 10A). Of note, among these nine genes, seven were fibroblast-enriched genes, indicating legitimate reprogramming to pluripotency, although insufficient (Figure 10C). In summary, OSKM reprogramming in an independent fibroblast line is predominantly legitimate, and ribosome biogenesis is globally reprogrammed to the pluripotent state.

## 3. Discussion

Several laboratories dissected the molecular events of iPSC reprogramming using genome-wide approaches such as microarray, RNA-seq, ChIP-on-chip [9,10,12,43], and proteomics [44]. Those studies had a fundamental deficiency. The authors implicitly treated all the transcriptional responses as positive events of pluripotency reprogramming. This is logically not sound because only a rare population becomes reprogrammed and data for 99% of the starting cells represent noise. To address this fundamental deficiency in the previous research, the author developed the concept of reprogramome [13]. Reprogramome provides a foundation for evaluation of reprogramming legitimacy [8]. This study first defined the subreprogramome for ribosome biogenesis, and then discovered that ribosome biogenesis was quickly and properly reprogrammed by the conventional Yamanaka factors by evaluation of the reprogramming legitimacy of all the 291 human genes of ribosome biogenesis.

This report established that human PSCs have a more robust ribosome biogenesis than the reprogramming starting cells, fibroblasts. To a scientist not specialized in ribosome biogenesis, this at a first glance seems a surprise considering that the house-keeping protein translational machinery is ubiquitous and essential for every cell. In fact, new information indicates that different tissues/cells may have a very wide range of concentrations of ribosomes that may vary by a factor of 3 to 10 [45]. Accumulating evidence indicates that a more robust ribosome biogenesis is a feature for growing cells, stem cells, and cancer cells. It is well known that the number of ribosomes in yeast cells is proportional to the rate of growth [17]. Drosophila germline stem cells demonstrate higher rRNA transcription [46]. Hyperactive ribosome biogenesis is a hallmark of cancer cells and has been an established diagnostic marker for cancers [14].

The current research agrees with many previous observations that components of the pluripotency ribosome biogenesis are more robust compared to somatic cells. Woonough et al. reported that rRNA synthesis decreased by 50% quickly upon directed differentiation of human ESCs [47]. Many genes of ribosome biogenesis have been reported to be highly expressed in pluripotent stem cells. FBL, a methyltransferase indispensable for rRNA processing, is highly expressed in mouse ESCs compared to the differentiated cells [23]. Nucleolin, involved in the early processing of the primary rRNA [48], is enriched in mouse ESCs (mESCs) and downregulated upon differentiation [49]. Many members of the SSU processome (SSUP) responsible for the generation of 18S rRNA are required for the maintenance of pluripotency, and downregulation of those SSUP components results in differentiation of mESCs [25]. Depletion of nucleolin in mESCs results in differentiation [49]. Ly1 antibody reactive (LYAR), involved in pre-rRNA processing [50], has the highest expression in mESCs compared to many cells including cancer cells and MEF, and is downregulated upon differentiation [26]. LYAR is critical for mESC self-renewal and differentiation [26]. The ribosome subunit components RPL7A, RPL36A, and RPS18, along with the new regulator of ribosome biogenesis HIV-1 Tat Specific Factor 1 (HTATSF1), are enriched in mESCs, and deficiency of HTATSF1 results in differentiation of mESCs [24]. The current research systematically demonstrated that ribosome biogenesis is globally enriched in human PSCs. The majority of previous observations are about mouse PSCs, whereas the current data represents discovery of a robust ribosome biogenesis in human PSCs.

This report, however, did find that 11 genes in ribosome biogenesis pathways had significantly higher expression in somatic cells. This may not contribute positively to ribosome biogenesis in fibroblast because of the following reasons. First, some of these 11 genes are negative regulators of ribosome biogenesis. Indeed, PTEN and KAT2B (also known as PCAF) are annotated as negative regulators of ribosome biogenesis. PTEN has a PI3K/Akt-independent function in nucleolus, and deficiency of PTEN has been found to enhance ribosome biogenesis in mammalian cells [51]. KAT2B negatively regulates ribosome biogenesis, likely by acetylation of PTEN, and promotes nuclear localization of PTEN [52]. In another case, KAT2B negatively regulated ribosome biogenesis by acetylating the component of the SSU processome U3-55K [53]. The acetylated U3-55K cannot bind to U3 snoRNA and therefore impairs processing of pre-rRNA to generate the mature 18S rRNA. In sum, the low expression of PTEN and KAT2B in hPSCs may further indicate a more robust ribosome biogenesis in human PSCs.

Another fibroblast-enriched gene annotated as “ribosome biogenesis” is *RNASEL*, which encodes the endoribonuclease L. However, there is no indication that ribonuclease L plays any role in ribosome biogenesis. The well-established role for ribonuclease L is to cleave viral and cellular RNA as the final executor in the OAS/ribonuclease L innate immune pathway in response to viral infection [11,54,55]. Ribonuclease L does cleave rRNA, but this is a step of apoptosis as a result of extreme cellular stress such as viral infection, and it is not a normal processing of rRNA for the maturation of rRNAs. In this case, ribonuclease L should also be a negative factor of ribosome biogenesis when ribosomes are destined for demolition in a cell undergoing apoptosis.

It is well known that some ribosomal proteins play extraribosomal roles. Some of these fibroblast-enriched genes in ribosome biogenesis may play roles other than ribosome biogenesis. For example, GTFIIH5 codes for the smallest subunit of the transcription factor II H (TFIIH) complex, p8, or TTDA. Although TFIIH plays a role in transcription of rRNA by Pol I [56], the prominent function of TFIIH is transcription of protein-coding genes by Pol II and the DNA repair pathway of nucleotide excision repair (NER) [57].

The report surprisingly demonstrated that OSKM quickly reprogrammed fibroblast ribosome biogenesis to the more robust pluripotent state in a very coordinated fashion. In alignment with the current data, previous research showed that some factors in ribosome biogenesis are required for iPSC reprogramming. For example, knockdown of the SSUP genes *Krr*, *Ddx47*, *Ddx52*, or *Pdcd11* all compromised mouse iPSC generation from MEF [25]. Overexpression of fibrillarin, a critical methyltransferase for rRNA processing, along with OSK in MEF increased number of iPSC colonies [23].

Of note, this report examined the impact of the four reprogramming factors as a group, as most previous studies have done, because no single reprogramming factor can convert fibroblasts into iPSCs. It will be interesting, in the future, to investigate the roles that each single reprogramming factor may play in pluripotency reprogramming. In terms of reprogramming of ribosome biogenesis, studying the role of an individual reprogramming factor may be more relevant since there is evidence that MYC is critical in regulating ribosome biogenesis [58]. The author’s lab is exploring whether MYC alone, or MYC along with another reprogramming factor, can quickly and globally reprogram ribosome biogenesis reported here.

MET was reported as an essential early event in pluripotency reprogramming from fibroblasts [9,30,31]. This report provided evidence that MET did not even initiate at the time of proper RB reprogramming. Not only did MET not occur at the time of proper reprogramming of ribosome biogenesis, but also there were all six types of aberrant reprogramming. Of note, the gene *MET* coding for the well-known EMT player, the receptor tyrosine kinase, was wrongly upregulated by 3.1-fold when it should be down-reprogrammed by 3.4-fold [59]. The current research provides convincing evidence that ribosome biogenesis represents the earliest major cellular process properly reprogrammed by the Yamanaka factors.

This report defines the subreprogramomes of ribosome biogenesis and MET for fibroblast reprogramming to human iPSCs, and demonstrates that RB was quickly reprogrammed prior to MET. The RB gene set was downloaded from the AmiGO2 database. The author noticed that some genes may not be annotated as ribosome biogenesis. The current annotations for human ribosome biogenesis may not be complete and 100% accurate. However, the incompleteness of the RB gene set does not change the conclusions drawn here considering the nature of the results. Using the same set of data and the same approaches, the conclusions about reprogramming of these two pathways are totally opposite, with one being properly reprogrammed and the other having no reprogramming. In each case, hundreds of genes behaved in the same ways, although there were some exceptions for a small group of genes. These two pathways served as intrinsic controls for each other. Within the RB pathways, there were also two completely different transcriptional responses to OSKM reprogramming—the hPSC-enriched RB genes were collectively upreprogrammed properly while the 96 shared RB genes were not impacted by the OSKM factors as they should have been. These two groups of genes within the RB pathways served as intrinsic controls to each other as well, indicating the high quality of the RNA-seq data and accurate reprogramming for RB pathways at the same time. Biased reprogramming for a specific pathway and for the hPSC-enriched gene set within a pathway also indicated that the conclusion of quick and authentic reprogramming is convincing because this large-scale biased reprogramming cannot be explained by random process. With more experimental evidence gained in the future, a more complete list of RB genes will someday allow for more thorough analyses.

Another limitation of the current research is that the RNA-seq was conducted on the poly(A)-purified population, and therefore the rRNA information is missing. However, a previous study indicated that rRNA is enriched by twofold compared to the differentiated cells [47], which agrees with the conclusion here that pluripotent RB system is more robust. Last, a more robust ribosome biogenesis was also observed in iPSC lines, and when compared to fibroblasts from a different donor. The quick and coordinated RB reprogramming was also observed in an unrelated fibroblast line.

## 4. Materials and Methods

### 4.1. Cell Lines and Cultures

The NIH-registered human embryonic stem cell (hESC) lines H1 and H9 were cultured in the chemically defined media as described previously [4,5,8,13]. Briefly, hESCs were cultured on Matrigel-coated vessels with E8 media [60], and passaged using EDTA-mediated dissociation when they reach 80% confluency. Human iPSC lines were culture in the same way as for hESCs.

Human primary fibroblasts (BJ, ATCC, Cat#, CRL-2522; CCD-1079Sk, ATCC, Cat#, CRL-2097) were culture in the fibroblast medium, Dulbecco’s modified Eagle’s medium, with high glucose, supplemented with 10% heat-inactivated fetal bovine serum, 0.1 mM 2-mercaptoethanol, 1× penicillin–streptomycin, 0.1 mM minimum essential medium non-essential amino acids, and 4 ng/mL human fibroblast growth factor 2 (FGF2).

### 4.2. iPSC Reprogramming from Human Fibroblasts

The reprogramming of human fibroblasts has been described previously [4,5,61]. Briefly, human fibroblasts were transduced with lentiviral reprogramming factors OCT4, SOX2, KLF4, and MYC at a multiplicity of infection (MOI) of 8:5:5:3. Viral particles were removed after overnight transduction by a medium change. The reprogramming cells were cultured in fibroblast media until harvest for RNA at 48 and 72 h post-transduction of the reprogramming factors.

### 4.3. RNA-seq

RNAs from the H1 and H9 ESCs were prepared independently on different days, and cells at different passage numbers for each ESC line were used to prepare RNAs of repeat samples. The repeat RNA samples for fibroblasts were prepared similarly to make sure the repeats were technically independent to each other. RNA-seq was conducted with the polyA+ population on the Illumina HiSeq2500 sequencer using the sequencing reagents and flow cells providing up to 300 Gb of sequence information per flow cell. The stranded mRNA library generation kits were used per the manufacturer’s instructions (Agilent, Santa Clara, CA, USA). The polyA mRNA was randomly fragmentized before cDNA preparation. cDNAs were generated using the random primers with inclusion of actinomycin D in the first strand reaction. The ends of the cDNA were repaired, A-tailed, and ligated with adaptors for indexing during the sequencing runs. We quantitated the cDNA libraries by qPCR before cluster generation. We generated approximately 725 to 825 K clusters per mm [2]. Cluster density and quality were determined during the run after the first base addition parameters were assessed. We conducted paired end 2 × 50 bp sequencing runs.

The more economic DNA nanoball (DNB)-seq technology was also used to RNA-sequenced 19 samples discussed in the last section. These RNA-seq comprised 4 groups with 4 biological replicates each and a fifth group with 3 biological replicates. Paired-end 100 bp reads were sequenced utilizing the DNBSEQ-G400 sequencing instrument at BGI.

### 4.4. Bioinformatics

Bioinformatics procedures have been reported previously [8,13]. Briefly, each RNA-seq has a minimum of 28.1 million reads, and the average number of reads are 40.1 million across all biological replicates. The FASTQ files were uploaded to the campus high performance computer cluster at the author’s university in order to analyze the RNA-seq data with the following custom pipeline built in the Snakemake system (v5.2.2) [62]: First, the author conducted quality control of the reads using FastQC, and trim the bases with quality scores of less than 20 using Trim_Galore! (v0.4.5). Following trimming, the author quasi-mapped the transcripts and quantified them with Salmon [63] to the hg38 human transcriptome from the Gencode release 29. The author then imported the quantification results onto a local RStudio session (R version 3.5.3), and the package “tximport” [64] was utilized for gene-level summarization. Finally, the author conducted differential expression analyses using DESeq2 package [65]. Boxplots were generated in RStudio using the published procedure by the author [66]. Heat maps were prepared in RStudio using the package *pheatmap* as reported [67]. All heat maps were prepared using log2-transformed data of the normalized read counts, and scaled by genes (row).

RNA-seq data was deposited with accession numbers of GSE148158 and GSE159410. RNA-seq data of 4 human iPSC lines—3RIPSC3 (GEO#, GSM1632433), 3RIPSC4 (GSM1632434), JQ1IPSC5 (GSM1953940), and JQ10IPSC (GSM2150917)—were used to further establish the fact that ribosome biogenesis is more robust in human pluripotent stem cells compared to fibroblasts. These 4 iPSC lines were established in the author’s laboratory and characterized as pluripotent [4,5].

### 4.5. Gene set of Human Ribosome Biogenesis

Human genes of ribosome biogenesis (GO #0042254) were downloaded from the Gene Ontology (GO) database AmiGO 2 by filtering the total annotations of GO #0042254 with “Homo Sapiens”. There are 316 genes with the 634 annotations as being human ribosome biogenesis. Of those, only 298 genes are available in our mapped RNA-seq data, with the remaining 18 genes as missing uncharacterized entries, most of which have no official gene symbols. This study concerned the more characterized 298 RB genes annotated in the database at the time of analyses (Appendix A).

### 4.6. Gene Set of Human MET/EMT-Related Genes

The complete list of 1184 human genes related to MET/EMT was downloaded from the EMT database, dbEMT2.0 [39]. This gene set was used to define the EMT/MET gene expression profiles in hPSCs and fibroblasts, and then the MET subreprogramomes for reprogramming of human fibroblasts into HiPSCs using the RNA-seq data of hESCs and fibroblasts.

### 4.7. Criteria for the Expressed Genes and Differentially Expressed Genes

RNA-seq data were normalized using *DESeq2* package on R. A gene is considered active only when the normalized read counts for all repeats are greater than 50. The rationale for these criteria has been reported [13]. A gene is regarded as inactive when the normalized read count for any of the repeat of a cell type or condition is <50 [8]. A gene was classified as differentially expressed when the fold difference is greater than 2 with a q value of less than 0.01, unless otherwise stated in the text.

## Figures and Tables

**Figure 1 cells-09-02484-f001:**
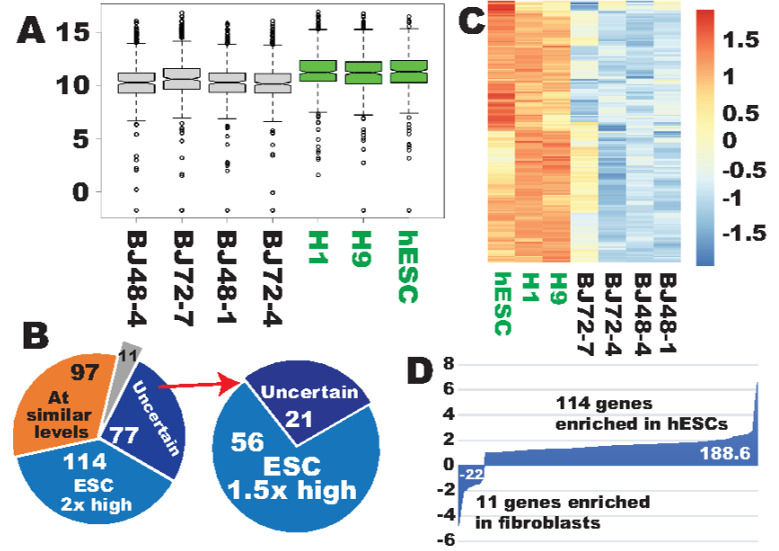
Ribosome biogenesis (GO#, 0042254) was more robust in human embryonic stem cells (ESCs) than in the reprogramming starting cells, fibroblasts. (**A**) Box plots showing that the full set of ribosome biogenesis (RB) genes was expressed at higher levels in human ESCs than in fibroblasts. (**B**) Pie charts showing the expression categories of the RB gene set in human ESCs (hESCs) and fibroblasts. The grey sector in the left pie chart indicates that 11 genes were enriched at least twofold in fibroblasts in comparison with human ESCs (q < 0.01). The right pie chart is a further classification of the uncertain sector in the left pie using less stringent criteria (>1.5×, q < 0.05). (**C**) A heat map showing that 170 out of 291 RB genes were enriched in human ESCs by at least 1.5-fold (q < 0.05). (**D**) Quantification of expression differences for the RB gene set between hESCs and fibroblasts. H1 and H9, human embryonic stem cell (hESC) lines H1 and H9, respectively; hESC is a subline of H1. The first number after the fibroblast designation is the time of RNA harvesting after seeding of cells, and the second number is the lane number of RNA-seq flow cells. The boxes and labels of hESCs are highlighted in green. hESC = 3; fibroblasts = 4. See also Appendix A.

**Figure 2 cells-09-02484-f002:**
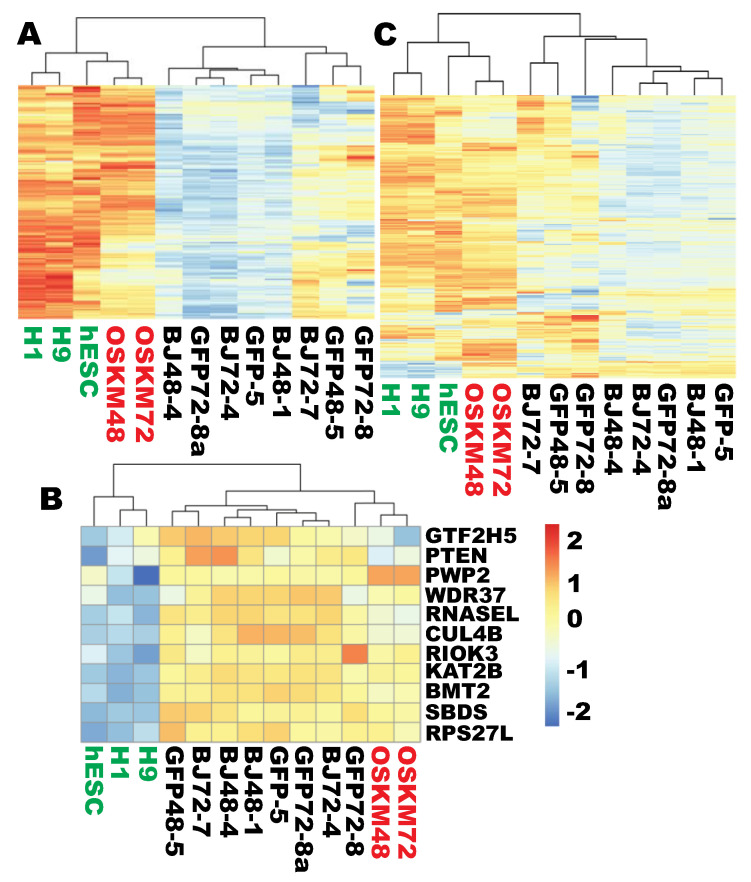
Proper reprogramming of ribosome biogenesis at the early stage of Yamanaka reprogramming. (**A**) A heat map showing that the 170 hPSC-enriched (>1.5×, q < 0.05) RB genes were upreprogrammed to the pluripotent state within 48 h. (**B**) A heat map showing that the 11 fibroblast-enriched RB genes were not downreprogrammed to the pluripotent state up to 72 h of OSKM induction and remained clustered with fibroblasts. (**C**) A heat map showing that the entire set of 298 RB genes became clustered with the hPSCs upon OSKM reprogramming. H1 and H9, human embryonic stem cell lines H1 and H9, respectively. The first number after the fibroblast or treatment designations is the time of RNA harvesting after seeding of cells or treatment time by OSKM or GFP, and the second number is the lane number of RNA-seq flow cells. hESCs are highlighted in green while OSKM samples are in red. hESC = 3; fibroblasts = 4; OSKM = 2. See also Appendix A.

**Figure 3 cells-09-02484-f003:**
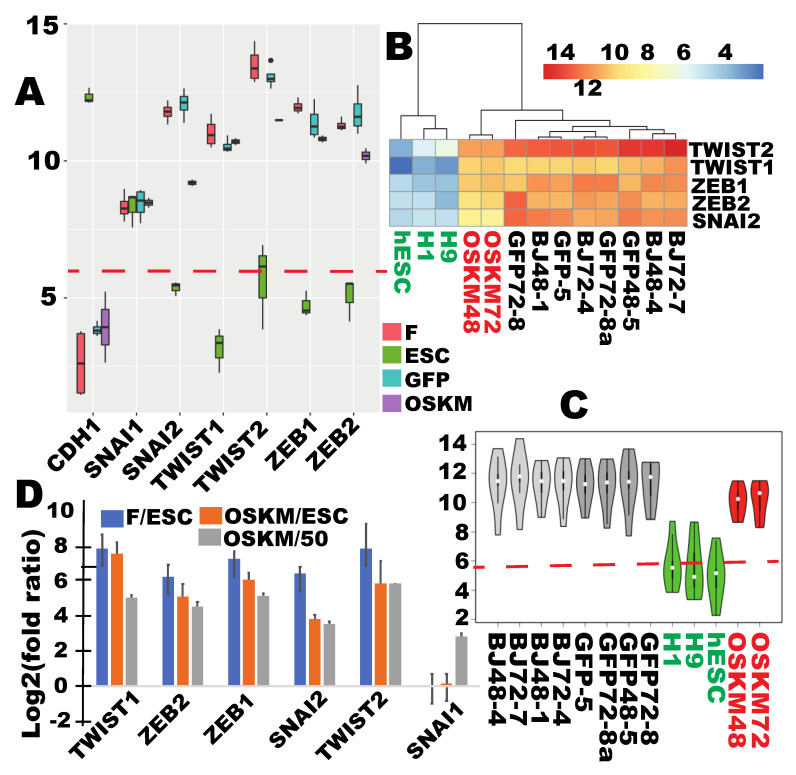
The key mesenchymal transcriptional factors (TFs) maintained high levels of expressions and the epithelial marker *CDH1* remained silenced at the time of proper RB reprogramming. (**A**) Box plots for individual genes showing little reprogramming of mesenchymal TFs, and of the epithelial marker gene *CDH1* at the early time of OSKM reprogramming (log2-transformed averaged read counts for time points of 48 and 72 h). (**B**) Clustering and the associated heat map showing lack of substantial downreprogramming of the five mesenchymal TFs. (**C**) Violin plots of each sample for the six mesenchymal TFs showing little reprogramming as a group upon OSKM induction. (**D**) Log2 (fold differences) relative to ESCs (blue and orange bars) or to the threshold level read counts (50) for the expressed genes (grey bars). F in (**A**) and (**D**) stands for fibroblasts. Dashed red lines in (**A**) and (**C**) mark the threshold level of the expressed genes. Sample labels are the same as in Figure 2. See also Appendix A.

**Figure 4 cells-09-02484-f004:**
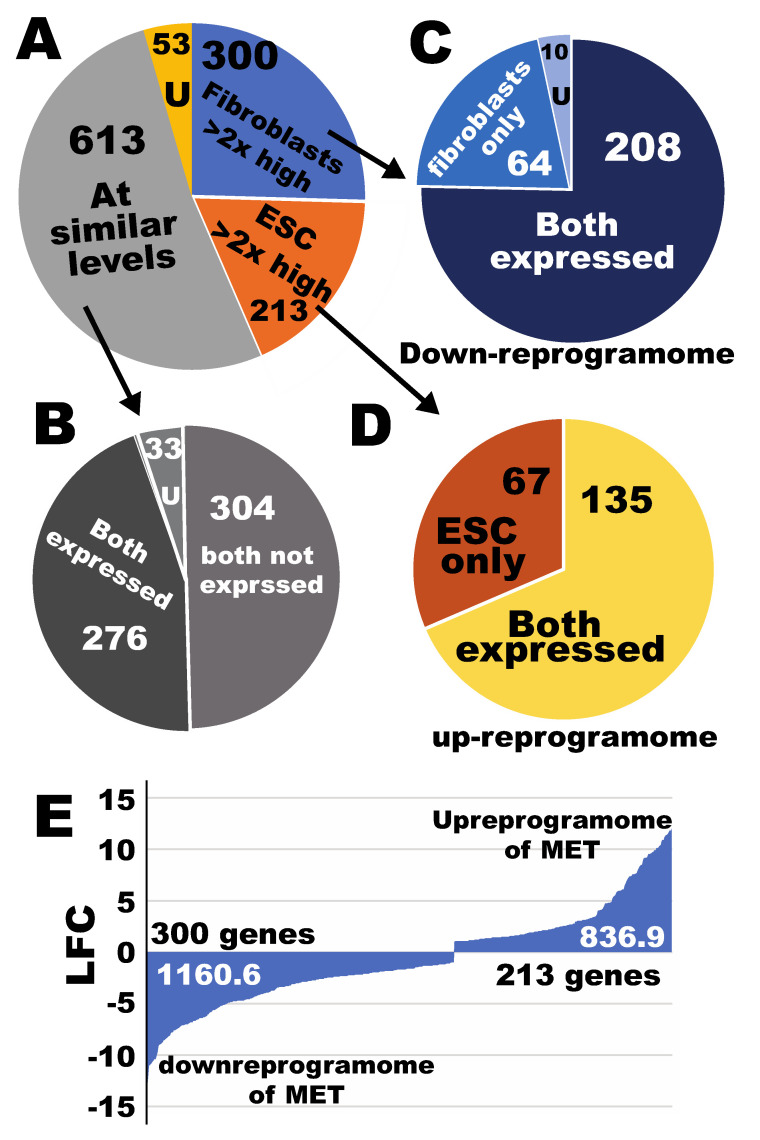
Defining the subreprogramomes for the mesenchymal-to-epithelial (MET) transition during the iPSC reprogramming process from human fibroblasts. (**A**) A pie chart for classification of the MET/EMT genes in the context of human fibroblast-to-iPSC reprogramming. (**B**) sub-classification for the sector of “at similar levels” in (**A**). (**C**) Sub-classification for the sector of fibroblast-enriched genes in pie chart (**A**). (**D**) Sub-classification for the sector of hPSC-enriched genes in pie chart (**A**). (**E**) Quantification of reprogrammikng for the MET subreprogramomes. LFC, log2 (fold changes); U, uncertain genes based on the sorting criteria. See also Appendix A.

**Figure 5 cells-09-02484-f005:**
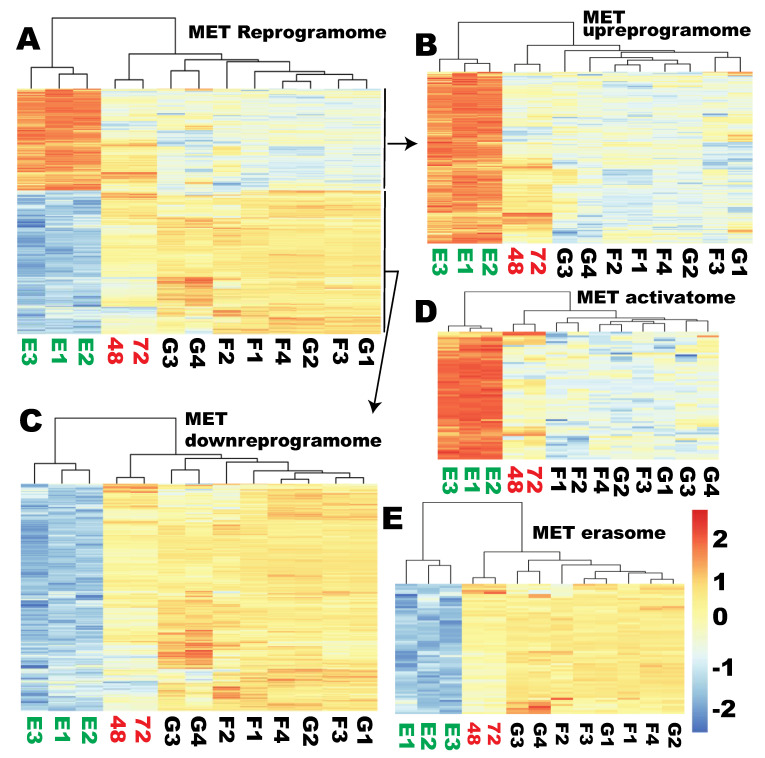
MET reprogramome and its subreprogramomes were not reprogrammed at the time of proper RB reprogramming. Clustering analyses and associated heat maps showing similarity of the reprogramming cells to the starting fibroblasts and GFP-transduced fibroblast controls and dissimilarity to hESCs for the entire MET reprogramome (**A**), upreprogramome (**B**), downreprogramome (**C**), activatome (**D**), and erasome (**E**). G1 to G4: GFP-5, GFP72-8a, GFP48-5, and GFP72-8; F1 to F4: BJ48-4, BJ72-7, BJ48-1, and BJ72-4; E1 to E3: human embryonic stem cell H1, H9, and hESC; 48 and 72: RNA harvested from human fibroblasts 48 and 72 h post transduction with lentiviral reprogramming factors OSKM.

**Figure 6 cells-09-02484-f006:**
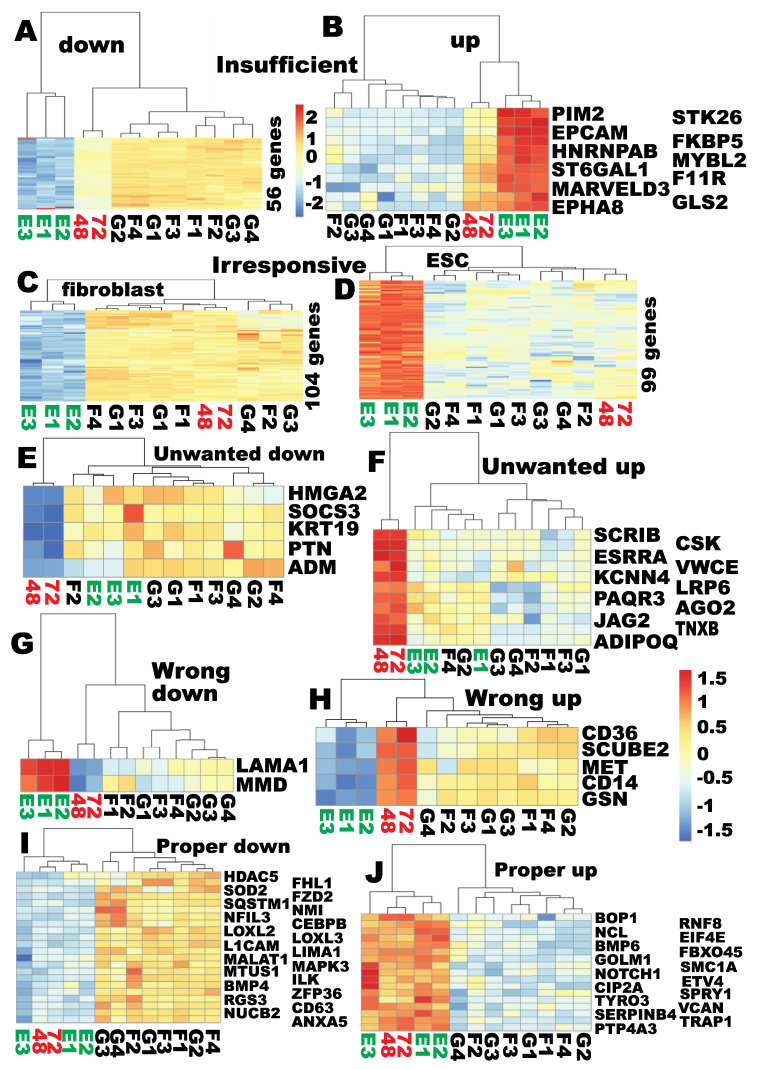
A Proper, Insufficient, Aberrant, and NO reprogramming (PIANO) response of the annotated MET genes to OSKM reprogramming at the time of proper RB reprogramming. (**A**) insufficient downreprogramming (56 genes); (**B**) insufficient upreprogramming (11 genes); (**C**) 104 fibroblast-enriched genes were irresponsive to OSKM reprogramming; (**D**) 99 hPSC-enriched genes were irresponsive to OSKM reprogramming; (**E**) 5 common MET genes were downregulated when they should have not been; (**F**) 11 common MET genes were upregulated when they should have not been; (**G**) 2 MET genes were wrongly downregulated when they should have been upreprogrammed; (**H**) 5 MET genes were wrongly upregulated when they should have been downreprogrammed; (**I**) 22 fibroblast-enriched genes were properly downreprogrammed; (**J**) 17 hPSC-enriched genes were properly upreprogrammed. Color scales of read counts are the same as shown by heat map (**A**), except for (**B**), (**G**), and (**H**) that share the second color scales as shown by (**H**). See also Appendix A.

**Figure 7 cells-09-02484-f007:**
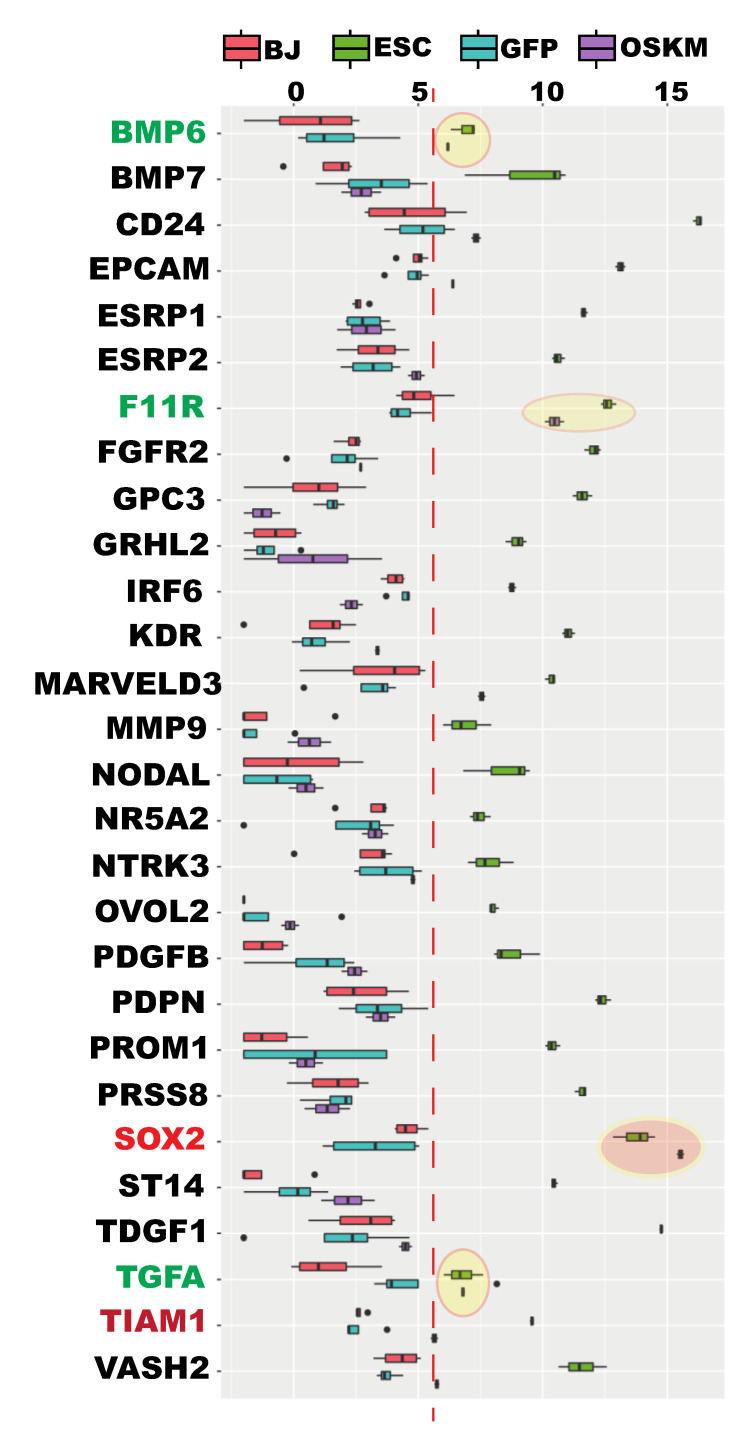
The hPSC-specific epithelial genes were not reprogrammed at the time of proper RB reprogramming. Box plots for each individual gene showing lack of reprogramming of the genes indicated, except for *SOX2*, *TGFA*, *BMP6*, and *F11R*. High level of *SOX2* was from the lentiviral transgene; *TGFA* and *BMP6* were properly reprogrammed to the pluripotency levels; *F11R* was upregulated significantly but insufficiently. See also Appendix A.

**Figure 8 cells-09-02484-f008:**
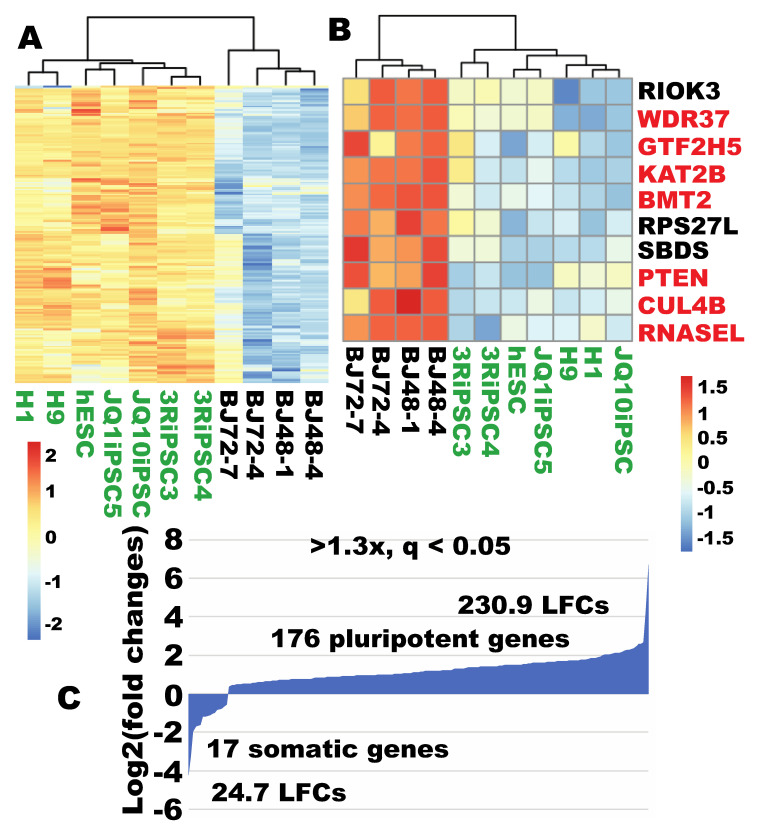
Ribosome biogenesis was more robust in human iPSCs than in fibroblasts. (**A**) Out of 298 RB genes, 176 were significantly enriched in human iPSCs compared to fibroblasts (>1.3×, q < 0.05). (**B**) Out of 298 RB genes, only 10 were enriched in human fibroblasts compared to iPSCs (>2×, q < 0.01). (**C**) Quantification of expression enrichment of the RB genes in iPSCs relative to fibroblasts. LFC, log2 fold changes.

**Figure 9 cells-09-02484-f009:**
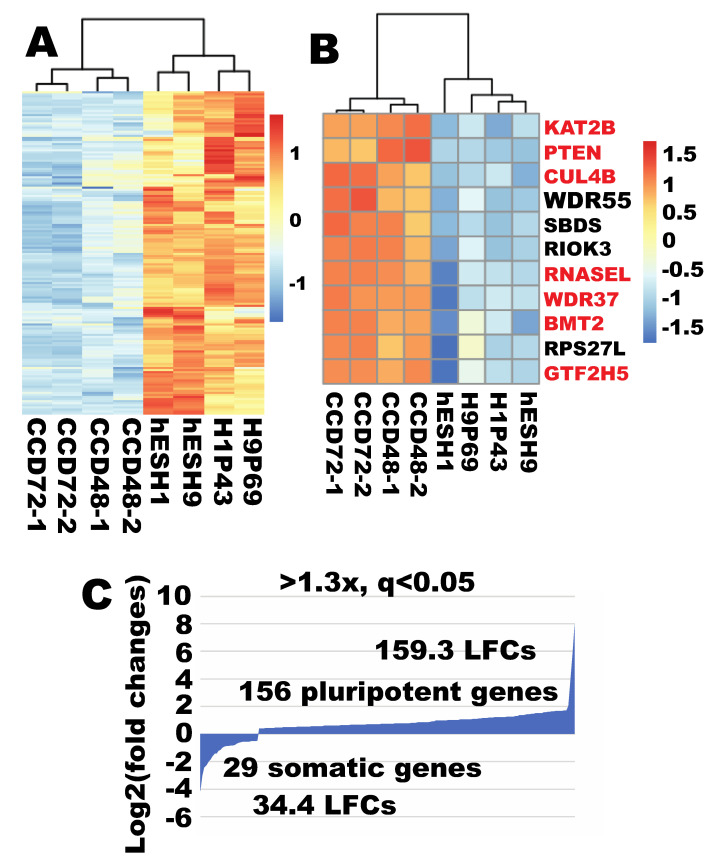
Ribosome biogenesis remained more robust in human ESCs compared with an independent fibroblast line. (**A**) Out of 298 RB genes, 156 were expressed significantly higher in ESCs (>1.3×, q < 0.05). (**B**) Out of 298 RB genes, only 11 RB genes were expressed significantly higher in fibroblasts than in ESCs (>2×, q < 0.01). (**C**) Quantification of expression enrichment of RB genes in ESCs relative to fibroblasts. H1P43 and hESH1 are human ESC line H1 at different passage numbers; H9P69 and hESH9 are human ESC line H9 at different passage numbers.

**Figure 10 cells-09-02484-f010:**
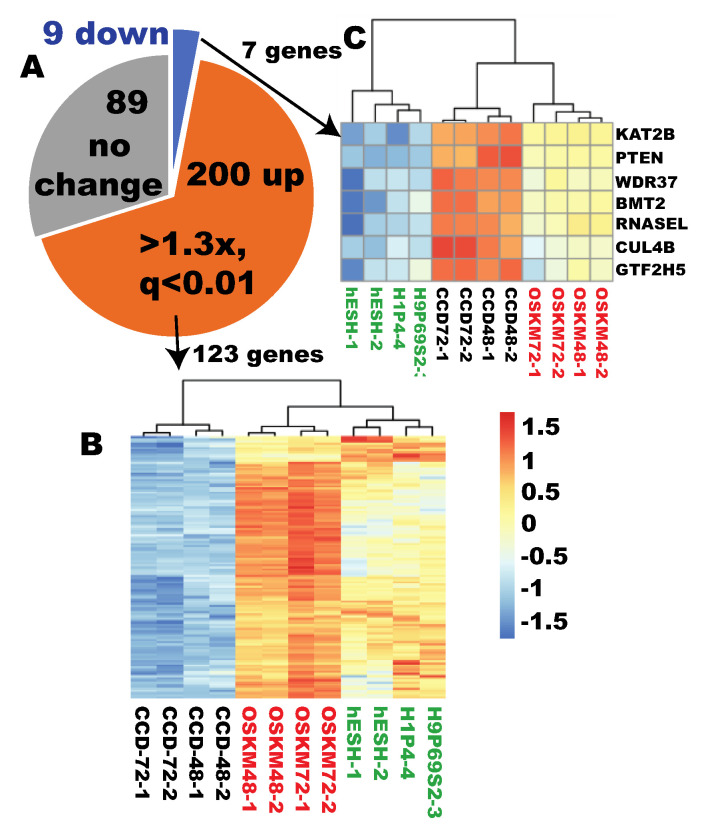
OSKM quickly reprogrammed RB to the pluripotent state in an independent fibroblast line. (**A**) Pie diagram showing predominant upregulation of RB genes during early iPSC reprogramming of the CCD fibroblasts. (**B**) Out of the 200 OSKM-upregulated genes, 123 were ESC-enriched (q < 0.05. >1.3×). Note the clustering of RB genes with ESCs and away from fibroblasts. (**C**) The fibroblast-enriched RB genes predominantly underwent legitimate downreprogramming, although insufficiently. Human ESC samples are indicated by green text; CCD fibroblasts underwent OSKM-mediated reprogramming are indicated by red color.

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
