# Peer review of "Quick, Coordinated and Authentic Reprogramming of Ribosome Biogenesis during iPSC Reprogramming"

_cells, 2020, doi:10.3390/cells9112484_

Round 1

Reviewer 1 Report

In this studyDr. Hu take into consideration the expression of ribosome biogenesis genes during iPSC reprogrammg by RNAseq. The presented results indicate that an intense reprogramming of ribosome biogenesis occurs during iPSC reprogramming.

My major concern regarding this study regards the fact that the contribution of Myc is not taken into consideration in the observed ribosome biogenesis reprogramming.

Myc is one of the master regulators of ribosome biogenesis promoting the transcriptional control of all three RNA polymerases and in particular of  RNA PolII on most of the genes involved in ribosome biogenesis. Myc is also one of the 4 genes overexpressed during iPSC reprogramming.   It is very suprising that this is not considered by the author: how many of the ribosome biogenesis factor reprogrammed in iPSC are known Myc target genes (or have an e-box consensus  sequence). I think that after the overexpression of myc it is not surprising to  observe an increase of its targets and if the reprogramming of ribosome biogenesis is limited to the overexpression of myc target genes then this should be carefully analysed in the results and extensively discussed.

In addition, the Author state the aim of the study at the beginning of the results (line 119-120), while in this reviewer's opinion this should find place in the introduction section.

minor issues:

figure 1 panel B: equal expression

line 215 - reprogramomes: the Author use this term and then explain its meaning later on (line 320) -I think this explanation should be anticipated

line 456 457. although ribosome biogenesis is a housekeeping process its quantitative regulation during proliferation is an extremely well known event, therefore I think I think its reprogramming in iPSC compared to fibroblasts is absolutely not surprising. 

Author Response

Dear Reviewers,

Thank you very much for your evaluation. I have made many major changes as suggested including but not limited to the following. Please note that all changes in the text were highlighted in red.

  1. Four RNA-seq data from four indepedent iPSC lines were included.
  2. Four RNA-seq data from four samples of an independent human fibroblast line were included.
  3. Four RNA-seq data from four OSKM reprogramming cells of an independent fibroblast line were included.
  4. Four new RNA-seq data from four samples of human ESCs were included.
  5. I removed all the ladder plots from 16 figures except for two.
  6. I re-draw all box plots and add color codes to box plots so that different cell types and treatments can be readily recognized.

Thank you for your further review. The following are point-by-point responses to your questions (response to reviewer 2 is in a separate sheet since Cells system requires response to each reviewer as a different file). Your original questions are enclosed in quotation marks.

“My major concern regarding this study regards the fact that the contribution of Myc is not taken into consideration in the observed ribosome biogenesis reprogramming.”

This is a great question. Most, if not all, of the previous studies iPSC reprogramming dealt with the four reprogramming factors as a whole because no single factor can convert fibroblast into iPSCs. I followed the conversion in my analyses. I strongly agree with the reviewer that the role of individual factor can be studied. This will be an independent project my lab will pursue. I would like to communicate the current findings first so that the scientific community will benefit from my findings, and science can mover forward faster. Sciences take more than one articles, and more than one labs.

“the Author state the aim of the study at the beginning of the results (line 119-120), while in this reviewer's opinion this should find place in the introduction section.”

I have shortened this statement, but still keep the major message so that readers can readily understand why I suddenly did such an analysis.

“figure 1 panel B: equal expression”

I have changed “equal expression” to “No difference in expression”.

“ine 215 - reprogramomes: the Author use this term and then explain its meaning later on (line 320) -I think this explanation should be anticipated”.

“Reprogramome” is defined in the introduction. Reviewer 2 think giving definition for reprogramome in line 320 is redundant with Introduction. Therefore, I deleted the paragraph of line 320.

“Line 456 457. although ribosome biogenesis is a housekeeping process its quantitative regulation during proliferation is an extremely well known event, therefore I think I think its reprogramming in iPSC compared to fibroblasts is absolutely not surprising. “

I have modified this sentence, which is highlighted in red.

Reviewer 2 Report

This study from Hu focuses on ribosome biogenesis (RB) during iPSC lentiviral reprogramming through ectopic expression of OCT4, SOX2, KLF4 and MYC (OSKM). First, by analysing RNA-seq data, it shows a global relative transcriptional enrichment of the RB related genes in ESCs compared to terminally differentiated fibroblasts. Next, it characterizes the early transcriptional response to OSKM, of these RB related genes. This analysis shows their reprogramming as soon as 48 hours after lentiviral treatment, to similar levels as ESCs. Finally, it reports that mesenchymal-to-epithelial (MET) transition, a well described early event of reprogramming, is not started yet at the same time when RB reprogramming is already achieved. Even though an accurate molecular understanding of the early events of iPSC reprogramming is surely of interest to the field and important for future technologies aimed to improve this inefficient process, the manuscript in its present form, does not present substantial conceptual novelty compared to the available literature. In addition, while some observations and conclusions proposed here are intriguing, they are based only on the analysis of a single set of RNA-seq and would require a more robust validation. Major concerns: 1) While an in depth characterization of the RB related genes function during reprogramming is relevant, the concept that RB is reinforced in pluripotent stem cells and may favour reprogramming is neither new nor unexpected. RB is reported to be “globally enriched” in stem cells compared to their differentiated counterparts (Stem Cell Reports. 2020 Sep 8:S2213-6711(20)30343-X, Cells. 2020 Feb 21;9(2):497, Stem Cells Int. 2020 Jul 6;2020:8863539, Curr Opin Genet Dev. 2015 Oct;34:61-70, Mech Ageing Dev. 2020 Jul;189:111282), ribosome incorporation promotes conversion of fibroblasts to multipotent cells (Sci Rep. 2018 Jan 26;8(1):1634), and, finally, MYC itself, which is a key constituent of the reprogramming OSKM cocktail, has been widely demonstrated to support RB (Nat Rev Cancer. 2010 Apr;10(4):301-9). The author himself extensively discusses the already known importance of RB for pluripotent cells (page 17, lines 464-481), underlining how this report is on human iPSC, while many previous findings were on mouse iPSC. The current manuscript would highly benefit from assessing at least one of the less understood and characterized key questions regarding RB and pluripotency/reprogramming: is the RB reprogramming observed here functionally relevant to promote reprogramming? Is it coupled to increased translation and protein synthesis? What’s the balance between these two processes? Is the RB reprogramming observed in response to OSKM, different from the ones observed when using other systems, which improve the reprogramming efficacy? As the data presented here, interestingly indicates that RB reprogramming is an early event that precedes MET, is this step required for the following MET? 2) As stated above, the accurate characterization of the RB related genes, which need to be up- and down-reprogrammed for an efficient reprogramming to take place, is of relevance for the field and may represent an important resource. However, the characterization presented here is based only on a set of RNA-seq from a single reprogramming experiment. As I can understand from the methods and the figures, only one biological replicate for each time point during reprogramming is provided, and compared to different ESC lines in single replicate. Why the author used duplicates for each fibroblast condition at a given time point, rather than 3 replicates? Why three different hESC lines were used, rather than, for example, biological triplicates of 2 lines? A more formally correct and better designed experimental plan will support the conclusions in a more robust way. At the very least, the author should take advantage of already published data from previous and independent transcriptional profiles during reprogramming, to validate their analyses and identify a shared gene signature of the RB, that can be generalized as an early feature of reprogramming. 3) Many of the conclusions in the present manuscript rely on analytical methods and mathematical models previously developed and reported by the same author (Int J Mol Sci. 2020 May 2;21(9):3229, Heliyon. 2020 May 27;6(5):e04035). While these tools may be useful to better define the transcriptional response to the OSKM and quantify the reprogramming, they were tested only on the same RNA-seq dataset commented before and it is not clear how generally applicable they are in the context of different reprogramming experiments. As they are linked to the wide concepts of “reprogramome” and “reprogramming legitimacy” by the author, the manuscript would benefit from applying these tools on other datasets. This will demonstrate that similar conclusions regarding RB reprogramming and MET, as the ones driven by the fig.1D, 4E and 6, are common features of reprogramming. Minor points: • Page 1, line 10: it is not clear what “evolutionary” means here. What does it mean that induction of iPSC is evolutionary? • Page 1, lines 33-37 and page 17, lines 447-449: it is not clear why previous reports failed to characterize the transcriptional landscape of iPSC. While it is true that the reprogramming is highly inefficient, some excellent reports provide the characterization of the few 5-10% fully reprogrammed cells, along with intermediates or partially reprogrammed cells (Nature. 2008 Jul 3;454(7200):49-55 Cell . 2012 Dec 21;151(7):1617-32) • Pages 1-2, lines 38-52, page 10, lines 319-324 and page 17, lines 449-454: the two previous publications from the same author are highly self-referenced throughout the text. The concept of “reprogramome” is well explained in the introduction. There is no need to redundantly repeat it. • Many figures are redundant, and data presentation should be improved. For example, in figure 2, panels A and B are redundant indicating similar conclusions, just using two different cut-offs for the same analysis. Figures 2D, is summarizing data already presented in other panels of figure 2. In supplementary figure S1 the 3 panels are showing exactly the same data in 3 different ways, but this is not required. In particular panel C is useless, as no information on the single genes is provided, and similar conclusions can be obtained from the heatmap. The same is also true for supplementary figures S2, S3, S6, S8-S13. More in general, the draft would highly benefit from showing additional data from other analysis (other datasets), rather than plotting the same data redundantly in different ways. • The pie charts in figure 1A and figure 4 are confused and difficult to be interpreted. The colour code of the charts and the texts is meaningless. In figure 1B the two pie charts should be inverted, as the q < 0.01 analysis is presented before the q < 0.05 analysis. Why the dimensions of the pie charts in figure 4A-D are all different? It does not seem they are proportional to the numbers, so is there a meaning for this? Maybe, barplots with a clear colour code and legend may be easier to be read. In general, the presentation of the data should be improved for easier and more readable interpretation. • In figure S2A (and other heatmaps showing unchanged genes between different conditions), it is not clear how data scaling was performed. It seems here that the data were scaled by rows, rather than by columns as in other heatmaps. If this is the case, why the author changes the way they analyse the data among different subsets of genes? • While the H1 and H9 hESC lines are reported in the methods, it is not clear what the hESC label refers to in the figures.

Author Response

Dear Reviewers,

Thank you very much for your evaluation. I have made many major changes as suggested including but not limited to the following. Please note that all changes in the text were highlighted in red.

  1. Four RNA-seq data from four indepedent iPSC lines were included.
  2. Four RNA-seq data from four samples of an independent human fibroblast line were included.
  3. Four RNA-seq data from four OSKM reprogramming cells of an independent fibroblast line were included.
  4. Four new RNA-seq data from four samples of human ESCs were included.
  5. I removed all the ladder plots from 16 figures except for two.
  6. I re-draw all box plots and add color codes to box plots so that different cell types and treatments can be readily recognized.

Thank you for your further review. The following are point-by-point responses to your questions. Your original questions are enclosed in quotation marks.

“the characterization presented here is based only on a set of RNA-seq from a single reprogramming experiment.”

I further compared RNA-seq four iPSC lines that were established and tested in my lab. I also analyzed four RNA-seq data from an independent human fibroblast line, recently sequenced in my lab for other purposes. I also included four more RNA-seq from human ESCs. In addition, my lab RNA-seq four more RNA from OSKM reprogramming cells (2 at 48 hours, and 2 at 72 hours). All of these used another independent RNA-seq technology, Nanoball sequencing, which is much cheaper. Our previous RNA-seq is based on Illumina technology, which is more expensive. RNA-seq data were deposited in public database, and access numbers were added in the manuscript. New data with new cells, and more repeats gave us the same conclusion.

“it is not clear what “evolutionary” means here”.

I am sorry this is a typo. The “evolutionary” should be “revolutionary”. iPSC technology is not a small advancement (“evolutionary progress”). It is revolutionary (A transformational advancement).

“Page 1, lines 33-37 and page 17, lines 447-449: it is not clear why previous reports failed to characterize the transcriptional landscape of iPSC. While it is true that the reprogramming is highly inefficient, some excellent reports provide the characterization of the few 5-10% fully reprogrammed cells, along with intermediates or partially reprogrammed cells (Nature. 2008 Jul 3;454(7200):49-55 Cell . 2012 Dec 21;151(7):1617-32) “.

I maintain the original statement in the two places of the original manuscript in that iPSC reprogramming efficiency is low and previous similar study of early reprogramming cells is flawed by the overwhelming noise because around 99% of the cells do not go in the direction of pluripotency. The early studies did not clearly employ the concept of reprogramming legitimacy. First, the 5-10% reprogramming efficiency in the Nature article mentioned is not true. The authors estimate the reprogramming efficiency as 20% at day 16 by evaluating the percentage of SSEA1+ cells using flow cytometry. This is very misleading. iPSCs grow much faster than the starting fibroblasts. The reprogramming vessels were overwhelmed by the fast growing iPSC cells at day 16. Their reprogramming efficiency is far below 1.2% because using the true pluripotency marker Nanog-GFP, the authors showed that there is only 1.2% Nanog+ cells at day 16 evaluated by flow cytometry. The relative more accurate way to evaluate reprogramming efficiency is to count the number of NANOG+ colonies and divide this number by the number of the starting fibroblasts. Therefore, my claim that reprogramming efficiency is <1% is correct.  Second, because they did not use my concept of reprogramming legitimacy, they clearly made at least one apparent wrong conclusion. In the last paragraph of the first section of results, the author claimed that PODXL activation by the Yamanaka factors was not desired activation of somatic lineage markers. In fact, PODXL is highly expressed in pluripotent cells (both in human and mouse) although it more well known as a marker of kidney podocytes. In fact, the Cell article mentioned indicated that the Nature article profiled the transcripts of cells that failed to reprogram (See Figure S1D, the turquois triangles, and the text of the last sentence of page 1623 the Cell article). Therefore, what the authors reported largely represent the noises when they did not apply the concept of reprogramming legitimacy. The Cell article still did not clearly propose the concepts of reprogramming legitimacy. Even the purified SSEA1+ intermediates are not pure and cannot be reprogrammed efficiently. For example, the Cell article showed the purified SSEA1+ population from day 3 reprogramming cells can be reprogrammed at only <10% (see their Figure 1B). I developed the concepts of reprogramome and reprogramming legitimacy recently. With these concepts, I do not just report what happen during the inefficient reprogramming process, and I evaluate the reprogramming legitimacy of each reprogramming gene. As a result, I can make discovery reported in the current manuscript.

“Reprogramome is well explained in the introduction. There is no need to redundantly repeat it.”

Fixed. I removed one redundant paragraph in section 2.5. In the beginning of section 2.7, I also deleted the description about PIANO responses to reprogramming factors since there is similar description in the Introduction part.

The pie charts in figure 1A and figure 4 are confused and difficult to be interpreted. The colour code of the charts and the texts is meaningless. In figure 1B the two pie charts should be inverted, as the q < 0.01 analysis is presented before the q < 0.05 analysis.”

Both pie charts were modified. The locations of the pie charts in Figure 1B is reverted and rotated to make better presentation. The text colors now were reduced to black and white. Two colors are still used because the pie sectors have to use different background colors. Like boxplots all over the manuscript, this boxplot was re-drawn so that different cell types use different colors, and all ESCs use green color. The corresponding ESC labels in the figures all use green color to make the figure easier to comprehend.

“There are redundant in many figures. For examples, ladder plots provide the same information as heat maps”.

I have deleted Figure 2A. I have deleted a total of 29 of the ladder plots from 16 supplementary figures, and keep only one ladder plot.

“Why the dimensions of the pie charts in figure 4A-D are all different? It does not seem they are proportional to the numbers, so is there a meaning for this? “

There is no meaning in the pie chart sizes. I have made these charts to similar sizes during revision.

Maybe, boxplots with a clear colour code and legend may be easier to be read. In general, the presentation of the data should be improved for easier and more readable interpretation.”

All boxplots were re-drawn and color codes were introduced to distinguish different cell types and treatments.

“While an in depth characterization of the RB related genes function during reprogramming is relevant, the concept that RB is reinforced in pluripotent stem cells and may favour reprogramming is neither new nor unexpected. RB is reported to be “globally enriched” in stem cells compared to their differentiated counterparts (Stem Cell Reports. 2020 Sep 8:S2213-6711(20)30343-X, Cells. 2020 Feb 21;9(2):497, Stem Cells Int. 2020 Jul 6;2020:8863539, Curr Opin Genet Dev. 2015 Oct;34:61-70, Mech Ageing Dev. 2020 Jul;189:111282), ribosome incorporation promotes conversion of fibroblasts to multipotent cells (Sci Rep. 2018 Jan 26;8(1):1634), and, finally, MYC itself, which is a key constituent of the reprogramming OSKM cocktail, has been widely demonstrated to support RB (Nat Rev Cancer. 2010 Apr;10(4):301-9). The author himself extensively discusses the already known importance of RB for pluripotent cells (page 17, lines 464-481), underlining how this report is on human iPSC, while many previous findings were on mouse iPSC. The current manuscript would highly benefit from assessing at least one of the less understood and characterized key questions regarding RB and pluripotency/reprogramming: is the RB reprogramming observed here functionally relevant to promote reprogramming? Is it coupled to increased translation and protein synthesis? What’s the balance between these two processes? Is the RB reprogramming observed in response to OSKM, different from the ones observed when using other systems, which improve the reprogramming efficacy? As the data presented here, interestingly indicates that RB reprogramming is an early event that precedes MET, is this step required for the following MET? “

Yes, literature indicates that ribosome biogenesis is more robust in stem cells, which was discussed in the current manuscript in detail. This supports my discovery, not dismiss my report. To the best of my knowledge, I am the first to systematically show that ribosome biogenesis is globally enriched in human pluripotent stem cells based on all the 298 RB proteins. Previously studies were limited in scale. I have this sentences in the discussion section: “The current research systematically demonstrated that ribosome biogenesis is globally enriched in human PSCs. Majority of previous observation is about mouse PSCs, the current data represents discovery of a robust ribosome biogenesis in human PSCs”.

This manuscript is also the first to demonstrate that Yamanaka factors quickly reprogram ribosome biogenesis to the pluripotent state in a global scale. I agree that my new discovery raises more questions. My lab will address further questions in our future research. Science takes more than one articles, and more than one lab to answer the questions. In fact, the importance of RB proteins in reprogramming was reported for some individual RB proteins, and these facts were discussed in the 7th paragraph of Discussion.

“What is the role of MYC in reprogramming ribosome biogenesis?”

This is a great question. Most if not all, of previous research about reprogramming dealt with the four reprogramming factors as a group, for example, MET transcription during early iPSC reprogramming. This is because no single factor can convert fibroblasts into iPSCs. I did the same in this manuscript. My lab has very strong interest to study the roles of individual reprogramming factors including MYC. We will do this in a more in depth and extensively way for the roles individual factors. I would ask the reviewers to allow dissemination of the current findings first so that the scientific community can benefit my discovery here, and the science can move forward faster as a whole. Science takes more than one articles, and more than one labs.

“While the H1 and H9 hESC lines are reported in the methods, it is not clear what the hESC label refers to in the figures”.

This one is a H1. The details about cell lines can be found in the GEO database using the accession numbers. I added some description about cell lines in the methods, and hESC is now clearly described as a subline of H1 in the legend of Figures 1 and S1.   All RNA samples for RNA-seq were prepared independently (on different days, with different passage numbers, and some samples were prepared several years apart. The added RAN-seq data during revision were samples prepared just recently this year.)

“In figure S2A (and other heatmaps showing unchanged genes between different conditions), it is not clear how data scaling was performed. It seems here that the data were scaled by rows, rather than by columns as in other heatmaps. If this is the case, why the author changes the way they analyse the data among different subsets of genes?”

In all the heat maps except for Figure S2A, the log2-transformed data were further scaled by genes (row) for high-resolution visualization. This information is added in the method section.  One scales the genes to prepare the heat maps as otherwise large expression values from highly expressed genes would dominate the plot. Heat map in S2A is different because these genes are a group of genes that are not differentially expressed. In this case, scaling of rows(genes) unnecessarily generate different colors even within the same cell type. Non-scaling and scaling of column better visualize the real difference of expression levels as revealed by statistic analyses. I used the ladder plot in this figure (Notes: other 28 ladder plots were deleted as reviewer 2 suggests since heatmap and ladder plots deliver the similar information).

Round 2

Reviewer 1 Report

In this revised version most of my previous comments were addressed (note in Fig 1D pie chart equal expression is still indicated as exression),

However my major concern relative to myc overexpression and its known role in ribosome biogenesis is bypassed by the author postponing its analysis to a later study. I can understand that the systematic analysis effect of myc would perhaps suggest to perform the same for all overexpressed factors thus widening the study beyond its initial scope (however this is what the results suggest. I request to include the issue at least in the discussion with a detailed and referenced argumentation of the topic.

Author Response

Dear reviewer 1,

Thank you for your further evaluation. As suggested, I added one paragraph in the Discussion seciotn. I also believe this added paragraph is a good in that the audiences are informed of the possible role of MYC in RB reprogramming.

I also did some further English refinement. All new revisions are highlighted in red except for the section titles which are intended to be red.

The following are my responses to your questions.

I appreciate your further help.     

Fig 1D pie chart equal expression is still indicated as expression”.

It was revised but “no difference in expression” was used. Now, the phrase of “at similar levels” is used to more accurately describe the situation. I now use “at similar levels” in the pie chart of Figure 4 as well.

However my major concern relative to myc overexpression and its known role in ribosome biogenesis is bypassed by the author postponing its analysis to a later study........ I request to include the issue at least in the discussion with a detailed and referenced argumentation of the topic”.

Thank you. I have added one such a paragraph in the discussions section. I also cited the key review reference about MYC regulation of ribosome biogenesis in that paragraph.

I agree that it is a great idea to study the roles of individual reprogramming factors. My lab will study this very soon since my postdocs are wrapping their current projects. We will conduct RNA-seq from samples when each individual reprogramming factors, all combinations of two factors, and all combination of three factors are over expressed. We will find out whether individual factor is exclusively responsible to the reprogramming of ribosome biogenesis reported here. If it is MYC (or MYC along with any other one), we will conduct MYC knockdown and conduct RNA-seq, conduct ChIP-seq of MYC, and other experiments. This will provide a in depth and detailed study.

Reviewer 2 Report

The author assessed most of the criticisms raised. The inclusion of new independent RNA-seq experiments improves the overall quality of the manuscript and supports more firmly the conclusions. As stated before, the idea of RB reprogramming as an early event of human iPSC reprogramming is interesting, and this new version of the manuscript may represent a resource to expand the research of these findings.  

Author Response

reviewer 2 has approved my revision. 

This manuscript is a resubmission of an earlier submission. The following is a list of the peer review reports and author responses from that submission.